# Speaking of the “Devil”: Diagnostic Errors in Interstitial Lung Diseases

**DOI:** 10.3390/jpm13111589

**Published:** 2023-11-10

**Authors:** Raluca Ioana Arcana, Radu Adrian Crișan-Dabija, Bogdan Caba, Alexandra-Simona Zamfir, Tudor Andrei Cernomaz, Andreea Zabara-Antal, Mihai Lucian Zabara, Ștefăniță Arcana, Dragoș Traian Marcu, Antigona Trofor

**Affiliations:** 1Doctoral School of the Faculty of Medicine, University of Medicine and Pharmacy “Grigore T. Popa”, 700115 Iasi, Romania; arcana.raluca@umfiasi.ro (R.I.A.); andreeazabara@yahoo.com (A.Z.-A.); stefanita-m-arcana@d.umfiasi.ro (Ș.A.); 2Clinical Hospital of Pulmonary Diseases, 700115 Iasi, Romania; radu.dabija@umfiasi.ro (R.A.C.-D.); simona-zamfir@umfiasi.ro (A.-S.Z.); dragos.marcu11@yahoo.com (D.T.M.); antigona.trofor@umfiasi.ro (A.T.); 3Department of Medical Sciences III, Pulmonology, Faculty of Medicine, University of Medicine and Pharmacy “Grigore T. Popa”, 700115 Iasi, Romania; tudor.cernomaz@umfiasi.ro; 4Department of Biomedical Sciences, Faculty of Medical Bioengineering, University of Medicine and Pharmacy “Grigore T. Popa”, 700115 Iasi, Romania; 5Regional Institute of Oncology, 700483 Iasi, Romania; 6Department of Surgery, Faculty of Medicine, University of Medicine and Pharmacy “Grigore T. Popa”, 700115 Iasi, Romania; mihai-lucian.zabara@umfiasi.ro; 7St. Spiridon Emergency Hospital, Clinic of Surgery (II), 700111 Iasi, Romania; 8Department of Medical Sciences I, Cardiology, Faculty of Medicine, University of Medicine and Pharmacy “Grigore T. Popa”, 700115 Iasi, Romania

**Keywords:** interstitial lung disease, diagnostic errors, diagnostic delay, pulmonary fibrosis

## Abstract

Interstitial lung diseases are respiratory diseases, which affect the normal lung parenchyma and can lead to significant pulmonary fibrosis, chronic respiratory failure, pulmonary hypertension, and ultimately death. Reuniting more than 200 entities, interstitial lung diseases pose a significant challenge to the clinician, as they represent rare diseases with vague and insidious respiratory symptoms. As such, there are many diagnostic errors that can appear along the journey of the patient with ILD, which leads to significant delays with implications for the prognosis and the quality of life of the patient.

## 1. Introduction

The preoccupation with unintentional medical errors is not new in the world, especially in the academic community. The study of imperfections in various health systems has led to worrying conclusions regarding the diagnostic error and subsequently the unintentional medical error as an undesirable consequence of excessive proceduralization, gross monetization, and protocol limitation of the medical act, which, no matter how much we try to template it, still remains at the discretion of a doctor’s sublime imperfection.

The most important alarm signal was sounded in 1999 by Kohn, Corrigan, and Donaldson through the publication that changed everyone’s perception of the therapeutic act, “To Err is Human”, showing that, regardless of the interventions, 1 in 10 patients suffer due to unintentional medical errors [1].

Interstitial lung disease (ILD) represents an umbrella term that brings together more than 200 respiratory diseases that affect the normal lung parenchyma and are accompanied by significant morbidity and mortality. In order to confirm a diagnosis, a combination of criteria is required: clinical, radiological, and sometimes pathological [2]. Classified from a clinical, imaging, and histopathological point of view, the classification has undergone multiple changes over time, consistent with the understanding of the complex pathophysiological mechanisms of these entities [3]. The most frequent ILD types that the clinician may encounter are sarcoidosis, hypersensitivity pneumonitis, idiopathic pulmonary fibrosis, ILD secondary to connective tissue diseases, ILDs induced by drugs, and pneumoconiosis [4].

In recent years, an additional entity included in the group of ILDs emerged, namely diffuse interstitial lung diseases with a progressive fibrosing phenotype. This progressive fibrosing phenotype raises a series of additional problems, as it is associated with an accelerated decline of lung function with more severe symptoms and a lower quality of life but also with a more reserved prognosis. A representative model in this respect is idiopathic pulmonary fibrosis, which reflects the concept of the progressive, self-sustaining destruction of normal lung parenchyma [5].

Establishing a diagnosis with increased accuracy in an acceptable time to initiate effective treatment represents a challenge for clinicians. Establishing an early diagnosis represents for the patient, in most cases, a better efficacy of the treatment; it can sometimes reduce unnecessary investigations, but it can also mean a better prognosis for the patient [6]. Unfortunately, the vague respiratory symptoms, relatively rare pathology, lack of sufficient information on ILD, and difficult access to certain key investigations can represent a delay in establishing the correct diagnosis, which can have important consequences for patients [7].

Diagnostic errors, although rarely discussed in medical practice, represent an old but real problem in the medical system. Although we benefit from a major evolutionary leap in modern medicine from all points of view, diagnostic errors remain a possibility whenever we are faced with complex medical cases in which other factors, both medical and non-medical, can intervene to make such an error possible [8]. Considering the significant impact that ILDs with a progressive fibrosing phenotype have on mortality and morbidity, the present review aims to signal the difficulties and diagnostic errors that can occur in the journey of the patient with these rare pulmonary diseases.

## 2. Diagnosing ILDs 

When discussing a positive diagnosis of diffuse interstitial lung disease, there are a number of steps that must be taken by both the patient and the clinician to ensure increased diagnostic accuracy.

The evaluation of such a patient starts from a high degree of clinical suspicion based on a detailed history and a rigorous clinical examination. We can affirm the fact that there are two main steps in establishing a positive diagnosis: a first step, which consists in establishing the diagnosis of ILD and differentiating it from other respiratory or non-respiratory diseases, and a second step represented by the etiological diagnosis of ILD [9]. This second step requires a series of extensive investigations, such as laboratory analyses, immune markers, lung function tests, a chest HRCT exam, a fibrobronchoscopic examination, a bronchoalveolar lavage, and in certain situations, a lung biopsy. Of all these investigations, the imaging exploration—an HRCT examination of the chest—is the main piece for diagnosis, as it can often bring indispensable information for diagnosis [5].

Once the physician accumulates all the necessary information, the ideal course of action is that the final diagnostic decision should be established within a multidisciplinary team (MDT). Currently considered the gold standard in the diagnosis of ILD, an MDT is made up of a respiratory physician, radiologist, and pathologist, to which a thoracic surgeon or rheumatologist may be added [3]. Lately, the role of the rheumatologist in the MDT committee has been emphasized, especially if a systemic autoimmune rheumatological disease is suspected [10]. The essential role of the rheumatologist is emphasized by the fact that we discuss complex pathologies, often with insidious evolution and vague symptoms, where the rheumatologist’s input and the range of investigations he can suggest can make the difference between a correct diagnosis and an erroneous one [11].

## 3. Challenges and Errors in the Diagnosis of ILDs

We searched the PubMed database for studies that analyzed misdiagnosis and delays in interstitial lung diseases. For search terms, we used “interstitial lung disease”, “diagnostic delays”, and “diagnostic errors” in order to identify studies that had these search terms in their title or abstract. A total of 20 studies that described diagnostic errors and challenges in interstitial lung diseases were found relevant for our review.

When we talk about errors and difficulties in the diagnosis of patients with ILD, it is necessary to identify the level where the problem could start. Of course, each stage of diagnosis comes with its own challenges and problems. In this sense, a first diagnostic error can even occur on behalf of the patient. For example, nonspecific symptoms possibly falsely attributed to age or another pathology; difficult access or even a lack of access to specialized medical care; and ignoring symptoms are all elements that make the diagnosis difficult [6]. The constitutional differences of each individual patient, the level of perception of the severity of the symptoms, but also the fragility of each individual are factors that belong to the patient and that influence the diagnostic process [12]. The time from symptom onset to a final diagnosis can vary, as shown in a study by van der Sar et al., in which only 30% of the patients with pulmonary fibrosis had a final diagnosis within 3 months; however, 40.2% of the patients received a final diagnosis in a year or more [13].

The diagnostic journey of the patient with ILD starts, in most cases, with the general practitioner. This visit plays a crucial role in the diagnostic progress of such a patient. Clinical suspicion of a disease such as ILD by the physician may mean a shorter time to diagnosis. A Finnish study by Purokivi et al. found that the majority of referral letters, 59%, were from primary healthcare, with a mean time from symptom onset to referral of 1.5 years, the main reason being suspicion of ILD [14]. In order not to omit essential medical information, the general practitioner must perform a detailed history and a rigorous clinical examination and must not ignore certain signs and symptoms that could indicate an autoimmune etiology or a connective tissue disease [7].

On the other hand, because they are rare respiratory diseases with relatively nonspecific symptoms—cough, exertional dyspnea, and fatigue—it is sometimes difficult to raise the suspicion of ILD in a simple visit to the doctor [15]. A systematic review by Carvajalino et al. of studies about patients with progressive fibrosing interstitial lung disease found that 68.2 to 98% of the patients manifested breathlessness and 59 to 94% had cough. However, these are not the only symptoms that the study reported. Interestingly, patients also manifested depression, affecting between 10 and 49.2 of the patients; upper gastro-intestinal symptoms, such as gastroesophageal reflux, in 35.7–100% of the patients; sleep related symptoms; weight loss; and fatigue [12]. All of these show the multitude of nonspecific symptoms of patients with fibrotic interstitial lung disease, which can confuse the clinician [12].

Subsequently, problems may also arise when the patient is referred to the community hospital or local respiratory physician, as there is the possibility of little to no experience in the diagnosis of ILD secondary to a small number of cases, which may cause this diagnosis to be overlooked. On the other hand, limited access to certain investigations, such as HRCT, respiratory functional explorations, and autoimmune markers, but also the lack of a multidisciplinary team or a link with a center specialized in ILD, are reasons why a diagnostic error could occur [12,16].

Unfortunately, due to these causes, patients are rarely diagnosed early, requiring multiple visits to different doctors to reach a final diagnosis or to be referred to a specialized center [17]. A study by Collard et al., who evaluated 214 Danish patients diagnosed with idiopathic pulmonary fibrosis, highlighted the fact that 53.2% of them required a second opinion and 38% of them were evaluated by at least three doctors until the diagnosis of pulmonary fibrosis was established [18].

Another study conducted by Gadi et al., which was carried out on patients diagnosed with idiopathic pulmonary fibrosis, revealed an interesting trajectory of these patients; 58% of patients were initially evaluated multiple times by the general practitioner, most often by an internist or less often by a cardiologist, and the initial symptomatology was in many cases ignored, overlooked, or attributed to other diseases. Some of these patients presented an acute episode of the underlying disease, which resulted in a presentation to the emergency room, subsequent hospitalization, being taken over by a respiratory physician, and only then being diagnosed with idiopathic pulmonary fibrosis [19].

The trend of visiting multiple doctors of different specialties before being referred to a specialist to make a positive diagnosis of ILD continues for these patients. Before arriving at a center specialized in ILD, patients are seen multiple times by their general practitioner and later referred to a local hospital, from where they are then directed further to specialized centers in diagnosing ILDs [20].

However, problems can be encountered even at the level of a specialized ILD diagnostic center. A small number of such centers specialized in the diagnosis of interstitial lung diseases, which makes it difficult for patients to access them; the lack of effective, clear communication between the members of the medical team; but also the complexity of some cases, which raises difficulties in both diagnosis and management can lead to errors and difficulties, thus having a significant impact on the prognosis and the quality of life of the patient [6].

Another challenging issue in the diagnostic process of these patients is the need for extensive investigations in order to establish an accurate diagnosis. In this regard, there are a number of real and important issues to consider when discussing errors and delays in ILD diagnosis, such as an increased number of investigations, but also their complexity, the paucity of well-trained medical staff to correctly interpret the results, a long period between conducting an investigation and receiving a result, and the need to repeat some tests in certain cases or diagnoses [21,22].

We can also talk about a series of risk factors that could cause diagnostic confusion. For example, patients with associated coronary disease, diabetes, or gastroesophageal reflux disease go through a longer diagnostic process compared with those without these comorbidities [12]. A study by Farkas et al. based on the EMPIRE registry showed that more than half of the patients, 51.6%, had idiopathic fibrosis-associated comorbidities, the most frequent being cardiovascular diseases and arterial hypertension [23]. Another set of risk factors for delaying the correct diagnosis, according to studies, can be male sex and advanced age [20].

## 4. What Does a Diagnostic Error Mean for the Patient?

Establishing the wrong diagnosis represents for the patient, first of all, a delay in establishing the correct diagnosis and starting the appropriate treatment. Studies show a delay of more than 12 months before a positive diagnosis is made, but also that up to 55% of ILD patients are misdiagnosed. The vague and nonspecific symptomatology that patients with ILD may present can mislead the clinician, leading to misdiagnosis [24].

The studies carried out in this regard on patients diagnosed with ILD are not numerous but tend to be quite extensive. In 2018, the Intensity study was conducted on a total of 600 subjects, who were recruited to participate in an online survey with 40 questions regarding the diagnosis of patients with ILD. Almost all survey participants initially consulted a general practitioner; of these, 27.8% were referred to a specialist after the first visit, but 30.4% reported multiple visits to their general practitioner before being referred to a specialist. More than half of the evaluated subjects (55%) reported at least one misdiagnosis, and more than one third (38%) reported two misdiagnoses before the final diagnosis was made. Among those who were initially misdiagnosed, the median time between the initial misdiagnosis and the final correct diagnosis was 11 months [22].

Hoyer et al., in 2019, conducted a multicenter cohort study of 204 patients with idiopathic pulmonary fibrosis, which analyzed symptom onset, first contact with a primary care physician, first hospital contact, referral to an interstitial lung disease center, first visit at an ILD center, and final diagnosis. The median diagnosis delay was 2.1 years, which was attributed to patients, general practitioners, and community hospitals. Of the 204 patients, 20% reported three or more visits to their general practitioner before being referred to a further specialist. The majority of the patients who were referred to ILD centers were investigated beforehand in other community hospitals, and they were rarely referred directly from a general practitioner [20].

In 2020, Brereton et al. conducted a study that analyzed patients diagnosed with idiopathic pulmonary fibrosis, studying the time interval from primary care until the first contact with a respiratory center, the moment of starting antifibrotic treatment, and the date of death. The median time from primary care until patients were referred to a respiratory clinic was 47 days, with 290 days until referral and presentation to a specialist ILD clinic and 540 until the start of antifibrotic treatment [25].

The results of these three studies are quite clear, in the sense that a large proportion of patients experience a delay in diagnosis, often significant, but this delay can be due to any structure involved in the diagnostic process, starting from the patient and ending with the tertiary centers involved in the diagnosis. The rest of the smaller and older studies suggest the same results. This delay in diagnosis is real and can consist of shorter periods of time, which can extend to a median duration of 1–2 years [12,18,26].

There are a number of other diseases that tend to have the same symptoms, thus, possibly creating confusion in establishing the correct etiology. In the study by Hewson et al., the most frequent diagnostic confusions were with chronic heart failure and COPD. These confusions can be attributed to the nonspecific symptomatology with which the patients present, namely cough and exertional dyspnea [27].

This symptomatology tends to be ignored in most cases or attributed to smoking, aging, or respiratory infections such as tuberculosis. In countries where the incidence of tuberculosis has increased, the differential diagnosis between these two pathologies can sometimes become difficult as reported in a study by Akhter et al., in which 38% of ILD patients in the studied lot were misdiagnosed and treated for tuberculosis [28]. Diagnostic errors can also be made with other diseases, such as asthma, pneumonia, bronchitis, emphysema, or allergies, and an initial wrong diagnosis is, unfortunately, frequent [17,18,22]. Most studies of patients diagnosed with ILD reported similar diagnostic confusions, suggesting unanimity regarding the initial misdiagnosis, as illustrated in the Table 1 [19].

Moreover, the consequences of these misdiagnoses reflect the prescription of an ineffective and possibly harmful treatment for the patient. According to studies, patients who were diagnosed with other pathologies received treatment with systemic corticosteroids, antibiotics, combinations of bronchodilator and inhaled corticosteroids, proton pump inhibitors, or antiacid therapy. An interesting thing is the administration of specific inhaled therapy even in patients who did not meet the criteria for asthma or COPD [20,22].

All these things, which have the consequence of delaying an accurate diagnosis, present a series of repercussions for the patient with ILD. For example, patients with idiopathic pulmonary fibrosis who are diagnosed later have a poorer prognosis and an increased risk of death. At the same time, the longer the delay, the lower the chances of a lung transplant up to the point of this chance being completely eliminated [22,29]. Also, another study shows that the longer the delay, the higher the extent of lung fibrosis identified in these patients, which also means a poorer prognosis [30].

The same idea is supported by a study by Brereton et al., who observed that patients with suspected ILD who were referred by other practitioners in the first 12 months had relatively preserved FVC values, a better TLCO value, and an increased time to death compared with patients evaluated after 12–24 months or who were referred after 24 months [25].

## 5. What Solutions Exist and How Can We Improve the Diagnostic Process?

Diagnostic errors and difficulties in managing ILD pathology represent a real-world problem that must be addressed, even if we aim to “discuss rare diseases”. The diseases are rare, but the patients are many. The fact that we are able to identify the error, quantify it, and analyze it is an important first step that can lead to the establishment of measures to prevent these problems. We can talk about the education of both the patients and medical staff, as well as about programs and interventions aimed at reducing the incidence of diagnostic errors as represented in Table 2 [8,31,32].

Improving the diagnostic process and shortening the time needed to establish an accurate diagnosis can take place through making changes at a patient level, at a medical personnel level, at the level of facilities, and the population’s access to these facilities, but also through deepening clinical studies. These goals can be achieved through patients’ medical education—through presenting to the doctor when the symptoms first appear without ignoring them or attributing them to other things, through improving diagnostic techniques, but also through facilitating patients’ access to specialized diagnostic centers in ILD, improving the bureaucracy of referring patients to specialists [19].

The general practitioner has an important role in the management of these patients, often being the first contact the patient has with the health system. The general practitioner should perform a careful anamnesis of the patient’s symptoms, followed by a clinical examination, where the pulmonary auscultation must be included and taken into account. An eventual association of symptoms, such as dyspnea and treatment-refractory cough, with insidious progression, accompanied by Velcro crackling rales, should alert the physician to a possible interstitial lung disease [6,33].

Another critical point that could benefit from improvement is represented by a more effective collaboration between different medical specialties in order to refer patients as efficiently and quickly as possible to a competent specialist in establishing the diagnosis of ILD. Enacting efficient and concise communication and a professional medical link between secondary and tertiary centers, including medical centers in a physical or virtual multidisciplinary committee, are key points to consider in the management of this pathology [24,34,35].

The multidisciplinary committee represents the gold standard in the diagnosis of ILD, but this is not always feasible, the lack of experts in this field being one of the main reasons. Another reason is that local hospitals do not have ILD specialists and uniting a multidisciplinary committee can be difficult from a physical point of view, so a relevant solution to this problem is represented by digital platforms, which could facilitate the creation of such committees [11,36,37]. A study by Fujisawa et al. corroborated a digital platform where patients with suspected ILD were included along with clinical, radiological, and histological data in order to benefit from the opinion of an MDT. The members of the team could access the patient’s medical information and hold online sessions in order to establish the diagnosis or the next course of action [38].

Why is this MDT so important? Considering the relatively low availability of this type of multidisciplinary team, it can become frustrating for the clinician and their contribution to establishing the diagnosis. Studies show that the multidisciplinary committee is important for the diagnosis, for recommending the necessary investigations to formulate a diagnosis, and also for treatment [11,39]. A study carried out on 93 patients showed that, following the multidisciplinary team discussion (MDT), the diagnosis of ILD underwent changes in 53% of cases and among patients with unclassifiable disease, 71% of them received a specific diagnosis, which also had implications on the therapeutic approach [40]. Another study by De Sadeleer et al., who evaluated 938 patients with ILD analyzed by the MDT, observed that in 79.5% of cases, a diagnosis was established, in 41.9% of cases, the diagnosis was changed, and in 19.5% of the patients, a clear diagnosis could not be obtained and the patients were recommended for further investigations [40].

Another course of action could be represented by the implementation of screening programs for certain categories of ILDs, especially those with a progressive fibrosing phenotype. However, a cost-effective screening program would only be possible in a targeted, high-risk population [6]. For example, patients with ILD in whom the chest HRCT examination reveals a pattern of the usual interstitial pneumonia (UIP) type usually have a more reserved prognosis, similar to patients with idiopathic pulmonary fibrosis [6]. Patients with a family history of ILD with a progressive fibrosing phenotype should also not be ignored, as more than 20% of them may present imaging-detectable interstitial lung changes [6,41].

On the other hand, we can bring to light a number of predisposing factors for ILD, such as male sex, advanced age, and smoking. For patients already diagnosed with autoimmune diseases or connective tissue diseases, there are a number of biological markers that can alert the clinician to the occurrence of ILD; for rheumatoid arthritis, the presence of rheumatoid factor and anti-citrullinated peptide antibodies; for systemic sclerosis, the presence of anti-Scl-70; and for polymyositis and dermatomyositis, the presence of anti-synthetase antibodies, anti-EXOSC, and anti-IFIH1 antibodies. Perhaps these categories of patients should benefit from screening for ILD both at the time of diagnosis and during the course of the disease, although at the moment, there are insufficient data on the optimal time interval for screening [42,43].

Another way that could alert the specialist to the presence of an ILD could be a modified lung function test, namely the measurement of lung volumes, spirometry, and TLco test. The presence of a restrictive syndrome, i.e., decreased TLCO, could indicate the presence of an ILD, but although it has a high specificity, the sensitivity is low [44]. However, in these situations, if warranted, additional investigations should be recommended in order to confirm or exclude a potential interstitial lung disease.

## 6. Conclusions

Patients with ILDs will have a difficult diagnostic path, strewn with difficulties, delays in the correct diagnosis, and sometimes even errors leading to ineffective treatments. Diagnostic errors remain a challenge for the clinician, given the complexity of the pathology but also the fact that we are talking about rare diseases. 

The problems encountered on the diagnostic journey belong to multiple sides: patients, doctors, but also the healthcare system that includes them all. This tells us that we are facing a complex issue for which we must take action through identifying the problem, followed by developing and implementing programs, allocating resources, and educating patients and clinicians on interstitial lung diseases. Only through taking proactive measures can we minimize the errors and the challenges in the diagnosis of ILDs.

## Figures and Tables

**Table 1 jpm-13-01589-t001:** Diseases that can be confused with interstitial lung diseases [17,18,19,22,28].

Respiratory Diseases that Can Mimic ILD	Other Diseases That Can Mimic ILD
Chronic obstructive pulmonary disease—COPD	Chronic heart failure
Infectious respiratory diseases Tuberculosis Pneumonia	Allergies
Asthma	
Bronchitis	
Emphysema	

**Table 2 jpm-13-01589-t002:** Measures to improve the diagnostic process in ILD.

Education and Information	Improving Access to Healthcare	Research
	Improving the System of Primary Care	
Simple and concise useful medical information for the general public Awareness and medical training for the recognition/raising of suspected ILD diagnosis among general practitioners Information regarding proper referral of these patients to an ILD specialist Informing patients about the importance and necessity of conducting full investigations to establish the correct diagnosis	Facilitating the patient’s access to the investigations necessary to establish the diagnosis Establishing a clear and effective communication network between family doctor, pulmonologist, and ILD specialist Establishing experienced multidisciplinary committees to guide medical specialists in establishing a positive diagnosis Increasing the number of physicians familiar with the management of ILD in community hospitals Access to investigations, such as functional explorations, from community hospitals	Development of an algorithm to identify patients at risk for progressive fibrotic disease Making comparisons between diagnostic errors in PID compared with other respiratory pathologies Identifying the median time to diagnosis of patients with PID, as well as when diagnostic delays or diagnostic errors occurred

## Data Availability

No new data were created or analyzed in this study. Data sharing is not applicable to this article.

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
