# Peer review of "Speaking of the “Devil”: Diagnostic Errors in Interstitial Lung Diseases"

_jpm, 2023, doi:10.3390/jpm13111589_

Round 1

Reviewer 1 Report

Comments and Suggestions for Authors

The paper “Speaking of the “devil”: diagnostic errors in interstitial lung diseases” by Arcana et al is a review paper on an interesting topic.

1. The paper is extremely lengthy and should be shortened by 40%. This is possible by implementing tables for listing of errors/problems and so on.

2. Many more of the statements need to be referenced. There is a paucity of references throughout.

3. The authors should state their strategy how they identified papers reviewed.

4. As the authors state by themselves, there are not many papers dealing on this topic. By entering into pubmed “diagnostic errors and ILD” I retrieved 736 refs. All these should be considered and rated systematically. Currently this review cites 42 papers

5. In particular in section 3. There is a lot of wording without much data. I expect from a review on the topic a concise and concrete listing of the errors occurring and also possibly estimates of likelihood/frequency

6. Many of the statement are just too simple: e.g. “history, rigorous examination should be done”; instead specific findings and tricks, best with supporting images should be presented

7. Table 1 essentially lists all pneumologic diseases; should be more specific what are the problems that lead to misdiagnosis; is it only negligence of the physician?

8. Try to be more specifc and delete “commonplaces” which are scattered throughout the text.

Author Response

  1. Unfortunately the paper cannot be shortened due to the suggested word count which must be at 4000 words.
  2. 3 more references were added.
  3. A paragraph regarding this issue was added
  4. Papers regarding this subject meeting our criteria were reviewed and selected for the article
  5. Section 3 was improved with data.
  6. A table was added to section 5 to illustrate better how to solve diagnostic challenges
  7. These are all diseases that can be misinterpreted as ILD according to the papers cited and it is often due to the lack of experience in ILD

Reviewer 2 Report

Comments and Suggestions for Authors

Dear authors, thank you for your quite interesting paper, however many of the presented data are very well known. For me it' s necessary to describe ILD in children population (chILD) or describe that adult population is only discussed. In chILD multidisciplinary committees exist, not only in US but also in Europe. Also other diseases that in table 1 are important especially in neonates, infants and young children.

Another suggestion some of the sentences, e.g. page 2 line 45 is too long and not so simple for understanding. 

Author Response

Thank you very much! 

Added some data to section 3 and we inserted a table in section 5.

Round 2

Reviewer 2 Report

Comments and Suggestions for Authors

Dear authors,

thanks for the second version of the manuscript, however I'm disappointed that stil there is misunderstanding of my suggestions. 

Author Response

I am sorry if there were any misunderstandings.
